# Combining Preoperative Clinical and Imaging Characteristics to Predict MVI in Hepatitis B Virus-Related Combined Hepatocellular Carcinoma and Cholangiocarcinoma

**DOI:** 10.3390/jpm13020246

**Published:** 2023-01-29

**Authors:** Si-Si Huang, Meng-Xuan Zuo, Chuan-Miao Xie

**Affiliations:** 1Department of Radiology, State Key Laboratory of Oncology in South China, Sun Yat-sen University Cancer Center, Guangzhou 510060, China; 2Department of Minimal Invasive Intervention, State Key Laboratory of Oncology in South China, Sun Yat-sen University Cancer Center, Guangzhou 510060, China

**Keywords:** combined hepatocellular–cholangiocarcinoma, microvascular invasion, hepatitis B virus, prediction, preoperative

## Abstract

Background: Combined hepatocellular carcinoma and cholangiocarcinoma (cHCC-CCA) is a rare form of primary liver malignancy. Microvascular invasion (MVI) indicates poor postsurgical prognosis in cHCC-CCA. The objective of this study was to investigate preoperative predictors of MVI in hepatitis B virus (HBV) -related cHCC-CCA patients. Methods: A total of 69 HBV-infected patients with pathologically confirmed cHCC-CCA who underwent hepatectomy were included. Univariate and multivariate analyses were conducted to determine independent risk factors that were then incorporated into the predictive model associated with MVI. Receiver operating characteristic analysis was used to assess the predictive performance of the new model. Results: For the multivariate analysis, γ-glutamyl transpeptidase (OR, 3.69; *p* = 0.034), multiple nodules (OR, 4.41; *p* = 0.042) and peritumoral enhancement (OR, 6.16; *p* = 0.004) were independently associated with MVI. Active replication of HBV indicated by positive HBeAg showed no differences between MVI-positive and MVI-negative patients. The prediction score using the independent predictors achieved an area under the curve of 0.813 (95% CI 0.717–0.908). A significantly lower recurrence-free survival was observed in the high-risk group with a score of ≥1 (*p* < 0.001). Conclusion: γ-glutamyl transpeptidase, peritumoral enhancement and multiple nodules were independent preoperative predictors of MVI in HBV-related cHCC-CCA patients. The established prediction score demonstrated satisfactory performance in predicting MVI pre-operatively and may facilitate prognostic stratification.

## 1. Introduction

Combined hepatocellular carcinoma and cholangiocarcinoma (cHCC-CCA) is an uncommon type of primary liver malignancy. Due to the presence of both hepatocytic and cholangiocytic differentiations, cHCC-CCA has overlapping clinical, imaging, and histological features of hepatocellular carcinoma (HCC) and intrahepatic cholangiocarcinoma (ICC) [1]. The incidence of cHCC-CCA varies among heterogeneous reports and ranges from 0.4% to 14.2% of all primary liver malignancies, which is probably underestimated as the majority of patients are diagnosed by postoperative pathology and patients who do not receive surgery may have been misdiagnosed with HCC or ICC [2]. Although the specific mechanism of the development of cHCC-CCA remains unknown, viral hepatitis is one of the most widely reported risk factors of cHCC-CCA occurrence. In previous studies, hepatitis B virus (HBV) infection was observed in up to 91.1% of the cHCC-CCA patients in Chinese population [3]. 

cHCC-CCA is highly aggressive and has a poor prognosis comparable to that of ICC and significantly worse than that of HCC [4,5,6,7]. Currently, the only treatment option that may cure cHCC-CCA is surgical excision. However, a high rate of recurrence is observed despite curative resection, reaching 80% at 5 years [2]. Though the data supporting locoregional therapies for cHCC-CCA patients are limited, partial response rates have been observed and downstaging for surgical resection as well as a survival advantage may be possible. Thus, it is essential to stratify the risk of recurrence in cHCC-CCA patients. 

In recent years, an increasing number of studies have been devoted to investigating prognostic factors of cHCC-CCA. Risk factors associated with prognosis in patients with cHCC-CCA have been inconsistent across studies, due to its low incidence, different etiological background, and distinctive histopathological features. Microvascular invasion (MVI) refers to the presence of micrometastatic tumor emboli in the intrahepatic portal vein or hepatic vein branches, which is a potential source of intrahepatic metastasis and distant metastatic spread [8]. Numerous studies have shown that MVI is associated with adverse biological features of HCC and predicts early postoperative recurrence [8]. Accurate assessment of MVI allows physicians to select appropriate treatment strategies, including widening the resection margin and neoadjuvant therapy for resectable patients [9]. Previous studies have shown that MVI is also an important prognostic factor for overall survival (OS) and recurrence-free survival (RFS) in patients with cHCC-CCA [3,10,11,12]. However, MVI is a histological feature that cannot be obtained prior to resection, despite the presence of biopsy due to intratumoral heterogeneity [13].

Preoperative prediction of MVI in HCC patients has been a hot topic over the past decade. Several risk factors, including tumor size, tumor number, serum alpha-fetoprotein level, and peritumoral enhancement, have been reported as being related to MVI in HCC [14,15,16]. Moreover, HBV replication and related factors have been reported to play a role in contributing tumor invasiveness in HBV-related HCC patients [17,18]. However, to our knowledge, there are few reports regarding preoperative prediction of MVI in HBV-related HCC-CCA patients. In this study, we aimed to investigate the predictive value of preoperative clinical characteristics and imaging features for MVI in patients with HBV-related cHCC-CCA.

## 2. Materials and Methods

### 2.1. Study Population

This study was approved by the Sun Yat-sen University Cancer Center institutional review board (approval no. B2022-056-01). Patients with pathologically confirmed cHCC-CCA who were admitted to the Sun Yat-Sen University Cancer Center between May 2012 and August 2021 were retrospectively reviewed. Patients meeting the following criteria were included: (1) patients with pathologically confirmed cHCC-CCA after surgical resection; (2) blood index examination performed within 2 weeks prior to surgery; (3) preoperative imaging (magnetic resonance imaging (MRI) or computerized tomography (CT)) performed in our radiology unit within 30 days before hepatectomy. Exclusion criteria were: (1) preoperative therapy, such as neoadjuvant chemotherapy and transarterial chemoembolization; (2) other malignancies being present; (3) incomplete clinicopathologic data; (4) hepatitis C infection. Of the 85 patients identified after exclusion, 69 (81.2%) patients had positive serum HBsAg (Figure 1). 

### 2.2. Follow-Up

After surgical resection, patients should be followed up 1 month after surgery and then 3–6 months thereafter. Radiological examinations (CT or MRI) and routine blood sample tests including tumor markers were performed during the follow-up. Recurrence was diagnosed when a new local or distant metastatic lesion was found by imaging examination. The time interval between the date of surgery and the first detection of recurrence or the last follow-up was defined as RFS. 

### 2.3. Image Analysis

All CT and MR images were retrieved from a picture archiving and communication system (Centricity Radiology RA1000, GE Healthcare). The images were assessed by two radiologists independently, with discrepancies in interpretation being resolved by consensus. The following imaging features were investigated: (a) the diameter of the tumor; (b) tumor number (single or multinodular); (c) the shape of the tumor (globular, lobulated, or irregular); (d) rim arterial phase enhancement; (e) peritumoral enhancement (defined as the presence of enhancement adjacent to the tumor boundary in the arterial phase); (f) washout appearance (defined as nonperipheral washout); (g) delayed central enhancement; (h) enhancing capsule; (i) peritumoral bile duct dilatation. The LR-M category was assigned to nodules based on Liver Imaging Reporting and Data System (LI-RADS) version 2018. Analyses were only conducted on a dominant nodule in patients with multiple liver nodules.

### 2.4. Clinical Data and Pathological Evaluation

Clinical information was collected from the electronic medical records system. In addition to general information (sex and age), peripheral blood parameters (neutrophil-to-lymphocyte ratio (NLR) and platelets), liver function indexes (albumin (ALB), γ-glutamyl transpeptidase (GGT), total bilirubin (TBIL), alanine aminotransferase (ALT)), and tumor biomarkers (α-fetoprotein (AFP), cancer antigen 19-9 (CA19-9) and carcinoembryonic antigen (CEA)) were retrospectively collected. HBV-related factors including HBsAg, hepatitis B virus e antigen (HBeAg), hepatitis B virus e antibody (HBeAb) and hepatitis B core antibody (HBcAb) were tested in all patients before surgery. The presence of MVI was obtained from the pathological reports after surgery. MVI was defined as tumor emboli, identified only by microscopy, in the vascular lumen lined with endothelial cells.

### 2.5. Statistical Analysis

All statistical analyses were performed using R version 4.0.3 (www.r-project.org) and SPSS statistical software package (version 22.0; IBM, Armonk, NY, USA). Continuous variables, except age and AFP, were converted into categorical variables based on routine cutoff points in clinical application. As AFP ≥ 400 ng/mL has been reported to be associated with MVI in previous studies [19,20], the cutoff value for AFP was defined as 400 ng/mL. Continuous variables are presented as means ± standard deviations and compared by one-way testing. Categorical variables are presented as numbers with percentages, and were analyzed using chi-square or Fisher’s exact tests. Multivariate logistic regression analysis was performed to identify independent risk factors for MVI. For the logistic regression analyses, McFadden’s pseudo-R^2^ value was calculated to evaluate the goodness of fit. Then a scoring system was developed according to the β regression coefficients using the method previously described [21]. We defined the β regression coefficient of an independent factor as the constant of the scoring system, corresponding to 1 point. the scores assigned to the other risk factors were the quotient of their β regression coefficient divided by this constant. Each point was rounded to the nearest integer. The prediction score for each patient was then calculated by summing the scores of the independent predictors. The receiver operating characteristic (ROC) curves were plotted, and the diagnostic value of the model was evaluated by the area under the curve (AUC). Kaplan–Meier survival curves were plotted to assess RFS, and statistical differences in RFS between groups were compared using the log-rank test. A *p* value less than 0.05 indicated a significant difference.

## 3. Results

### 3.1. Clinical and Imaging Characteristics of Patients

A total of 69 patients with HBV-related cHCC-CCA were retrospectively enrolled in this study. Most patients were male (n = 56, 81.2%) and the median age was 52.0 years (IQR 46.0–57.0). Postoperative histopathological results revealed that 29 (42.2%) patients were MVI-positive. Table 1 shows the clinical and imaging characteristics stratified by MVI status. HBeAg was positive in 12 (17.4%) patients, and no differences were shown between MVI-positive and MVI-negative patients. A significantly higher serum level of γ-glutamyl transpeptidase (GGT), larger tumor size and higher proportion of multiple nodules were observed in MVI-positive group compared with MVI-negative group (*p* < 0.05). AFP ≥ 400 ng/mL was observed in 21.7% (15/69) of patients, showing no difference between MVI-positive and MVI-negative groups.

Peritumoral enhancement was the only imaging characteristic significantly associated with MVI (*p* = 0.001). 63.8% (44/69) of cHCC-CCA were classified as LR-M. This proportion was slightly higher in MVI-positive patients than that in MVI-negative patients, but no significant differences were observed (72.4% vs 57.5%, *p* = 0.308). Other imaging characteristics including tumor shape, rim enhancement, wash out, delayed central enhancement, peritumoral bile duct dilatation and enhancing capsule showed no difference between the two groups.

### 3.2. Univariate and Multivariate Analyses

Table 2 shows the results of univariate and multivariate logistic regression analysis of risk factors. There are four factors associated with MVI in univariate logistic regression analysis, including tumor size, multiple nodules, GGT and peritumoral enhancement. Multivariate analysis revealed that a higher level of GGT (OR, 3.69; 95% CI, 1.10–12.37; *p* = 0.034), peritumoral enhancement (OR, 6.16; 95% CI, 1.78–21.36; *p* = 0.004) and multiple nodules (OR, 4.41; 95% CI, 1.06–18.39; *p* = 0.042) were independent variables associated with MVI. The multivariate logistic regression yielded a McFadden’s pseudo-R^2^ of 0.46, suggesting a good fit.

### 3.3. Ability to Predict MVI

A scoring system incorporating the independent predictors was constructed according to the β regression coefficients derived from multivariate analysis (Table 2). The total prediction score for each patient was calculated as the sum of scores corresponding to the independent risk factors of this patient, yielding a range from 0 to 3. A higher score indicates a greater risk of MVI. The ROCs of the score and each predictor are reported in Figure 2. The AUC for the prediction score of MVI was 0.813 (95% CI 0.717–0.908), which was higher than that of each single predictor. The optimal cutoff score was 1, with a sensitivity of 93.1% and specificity of 55.0% for MVI. The presence of MVI was observed in 60% of high-risk patients (score ≥ 1; n = 45), and the prevalence of MVI was 8.3% in low-risk patients (score < 1; n = 24), exhibiting a definite difference (*p* < 0.001). 

After surgical resection, the median RFS for patients with MVI was 3.9 months, compared with 16.4 months for patients without MVI. Kaplan–Meier survival curves were plotted as shown in Figure 3A, and a significant difference was found in RFS between MVI-positive and MVI-negative patients (*p* < 0.001). In high-risk patients with a higher prediction score, the median RFS was 4.2 months, while low-risk patients had a median RFS of 27.4 months. Significantly better RFS was observed in patients with lower risk (*p* < 0.001; Figure 3B). 

## 4. Discussion

cHCC-CCA is a distinct subtype of primary liver malignancy, with overlapping features of HCC and ICC. The etiology and risk factors may be different between Eastern and Western countries [22]. HBV infection is the predominant etiological factor of cHCC-CCA in east Asia, resembling that in HCC [23]. Although the results of fundamental study suggested a potential role for HBV integration in tumorigenesis of cHCC-CCA, the association between HBV infection and development of cHCC-CCA remains unclear. Excluding potential impacts on results, we focused on HBV-related patients in this study. In our results, peritumoral enhancement, GGT and multiple nodules were demonstrated to be independent risk factors of MVI in HBV-related cHCC-CCA patients. The scoring model that we established combining the predictors exhibited satisfactory predictive accuracy. Additionally, the scoring model was able to successfully classify patients into high- and low-risk groups with significantly different RFS rates.

MVI, indicating aggressive biological behaviors of tumors, has been accepted as a vital risk factor for early recurrence after hepatectomy [8]. Previous studies have demonstrated that a wide resection margin significantly improves the RFS and OS in HCC patients with MVI [24]. Therefore, accurate preoperative estimation of MVI is helpful to select appropriate surgical modality, thus improving the prognosis. The incidence of MVI in patients with cHCC-CCA has been reported to vary from 30% to 68.7% [25]. In recent years, the postoperative prognostic significance of MVI in patients with cHCC-CCA has been clarified. Chu et al. reported that microvascular thrombus was one of the independent predictors for decreased OS in patients with HBV-related cHCC-CCA [3]. Wang et al. constructed a nomogram to predict postsurgical prognosis of cHCC-CCA and determined that MVI was independently associated with OS and RFS [26]. The same results were also revealed by zhang et al. in a research of 296 Allen type C cHCC-CCA patients and a prognostic scoring method incorporating MVI and other seven factors was established and outperformed the existing staging methods [27]. In the current study, we observed a significantly lower RFS in cHCC-CCA patients with MVI, consistent with previous studies. A scoring model was further constructed for preoperative prediction of MVI, which may provide a reference for clinical decision-making. However, more evidence is needed to clarify the survival benefit of individualized treatment strategies based on preoperative MVI assessment.

Previous studies demonstrated that the clinicopathologic features of cHCC-CC, especially HBV-related cHCC-CCA, resemble those of HCC more than ICC, while cHCC-CCA presents to be more aggressive than HCC and more similar to ICC [28,29]. This may relate to the distinct disease pathogenesis of cHCC-CCA. HBV X protein has been reported to be associated with the development of MVI in HCC patients [30]. Lei and Wei et al. analyzed the impact of the status of HBV infection in large cohort studies, and found that active replication of HBV was significantly associated with the presence of MVI in HBV-related HCC patients [18,31]. The result of our study demonstrated that active HBV replication, as indicated by HBeAg positivity, was not associated with MVI in HBV-related cHCC-CCA patients. The underlying mechanism of development of MVI in cHCC-CCA needs further study. AFP is another well-known risk factor for MVI in HCC [31,32]. In our results, no significant difference in serum AFP levels was observed between the MVI-positive and MVI-negative groups. However, Wang et al. demonstrated that AFP is an independent predictive biomarker for MVI in cHCC-CCA patients [19]. Another study showed that AFP was related to MVI in cHCC-CCA patients but was not an independent risk factor in multivariate analysis [20]. Apart from etiology, one possible reason for this discrepancy is that among patients with advanced fibrosis and chronic hepatitis, the serum AFP level can be elevated, so the sensitivity and specificity of AFP are limited [33,34].

Peritumoral enhancement has been widely accepted as an important risk factor associated with MVI [15,35]. This relates to the hemodynamic perfusion changes present in the area of decreased blood flow because of tumor thrombi in minute portal branches, which leads to compensatory arterial hyperperfusion [36,37]. Thus, it is not difficult to understand the utility of this imaging feature in the prediction of MVI in cHCC-CCA shown in our study, which is consistent with a previous study [19].

An interesting finding of this study was that higher serum GGT levels were significantly associated with MVI of HBV-related cHCC-CCA. GGT is a commonly used index of liver dysfunction. In the past decades, there have been important advances in the understanding of its physiological basis and its role in tumor biology. GGT is a membrane enzyme which catalyzes the degradation of extracellular glutathione (GSH) and provides precursors for the synthesis of intracellular GSH, the major endogenous antioxidant [38]. Increased GGT expression is common in human tumors [38]. It has been suggested that GGT expression may provide a growth advantage to tumor cells. One hypothesis is that GGT-mediated GSH metabolism maintains intracellular GSH levels, allowing cells to resist the toxicity of promoting agents and respond to proliferative signals triggered by oncogenic regimens [39]. In addition, this GGT-dependent process can increase cysteine supply for intracellular protein synthesis in rapidly dividing tumor cells [40,41]. Moreover, reactive oxygen species are produced as by-products of GGT-mediated GSH metabolism, which subsequently induce DNA damage and genome instability, and further promote cellular growth and proliferation [41,42]. Although the exact mechanism of GGT in the development and progression of cancer needs further investigation, numerous clinical studies have shown that elevated serum GGT levels are associated with poor prognosis of primary liver cancer, including HCC and ICC [42]. A retrospective study including 390 patients demonstrated that a higher serum GGT level was an independent predictor for worse OS and RFS in patients with HBV-related cHCC-CCA [3]. A higher level of GGT was demonstrated to be related to larger tumor size, vascular invasion and advanced BCLC stage in a previous study including 219 HBV-related HCC patients [43]. A study by Zhao et al. found that a higher serum GGT level (>130 U/L) was an independent risk factor for MVI in HCC patients [14]. Similarly, in a study of 714 non-metastatic HCC patients, a significant association was shown between serum GGT levels and the presence of MVI [44]. In a multicenter study, Chen et al. demonstrated that GGT was also an independent predictor for the occurrence of MVI in ICC patients [45]. This study is the first to demonstrate the predictive value of GGT for MVI in cHCC-CCA patients, and this finding needs to be further validated in a larger population.

Multinodular tumor was another independent predictor of MVI in cHCC-CCA patients, which has not been reported previously. However, in previous studies, multiple tumor nodules have been shown to correlate significantly with microvascular invasion in HCC patients [46,47,48] and may also be an independent predictor for MVI in ICC patients, which was previously identified by Chen et al. [45]. Previous studies have reported that tumor size is one of the most important risk factors for MVI in HCC [8,46]. However, there are conflicting results concerning the performance of tumor size in predicting MVI independently [48,49]. In this study, although univariate analysis revealed that tumor size was related to MVI of cHCC-CCA, tumor size was not an independent predictor of MVI in the multivariate analysis, consistent with previous studies [19,20].

Currently, there are no clear guidelines for the management of cHCC-CCA. Surgery is the only curative approach available for localized tumors. A major hepatic resection is recommended [50]. However, a high risk of early recurrence after surgery is present. Besides, the majority of HBV-related patients have underlying cirrhosis. In this case, patients need to be accurately selected for hepatectomy to avoid treatment-related complications such as bleeding and liver failure, and the volume of liver that can be removed is limited due to reduced liver function reserve [51]. Although the data on the benefit of adjuvant therapies, including transarterial (chemo) embolization, hepatic arterial infusional chemotherapy, systemic chemotherapy and transarterial radioembolization, are limited, partial responses were observed and may benefit cHCC-CCA patients in subsequent surgery and survival [52,53,54]. Thus, the identification of cHCC-CCA patients who would benefit from aggressive surgery and adjuvant therapies is needed. A preoperative scoring model was created in our study to predict MVI, and patients at high risk for MVI had a significantly lower RFS than patients at low risk. The prediction score is convenient to collect and use in routine clinical practice, and may aid clinicians in decision-making before surgery.

There are several limitations in this study. First, the present results were derived from a single-center and retrospective study, and our sample size was relatively small. Second, the current study may not be able to fully reveal the relationship between HBV status and MVI, since the replication status was assessed by HBeAg which is not as reliable as HBV-DNA. Third, this study proposed a novel scoring system which could be easily applied to clinical practice. The predictive performance shown in the results was acceptable but not excellent and the prediction score had the disadvantage of low specificity. Thus, further exploration of better predictors is needed to improve the predictive accuracy. Finally, due to the rarity of the disease, there was no validation group to verify our findings.

## 5. Conclusions

In summary, our results indicated that, in HBV-related cHCC-CCA patients, peritumoral enhancement, multiple nodules and GGT were independent risk factors for MVI. On this basis, we created a novel scoring method to predict the occurrence of MVI, which, if further confirmed in a study with a larger sample size, may facilitate clinical decision making before resection.

## Figures and Tables

**Figure 1 jpm-13-00246-f001:**
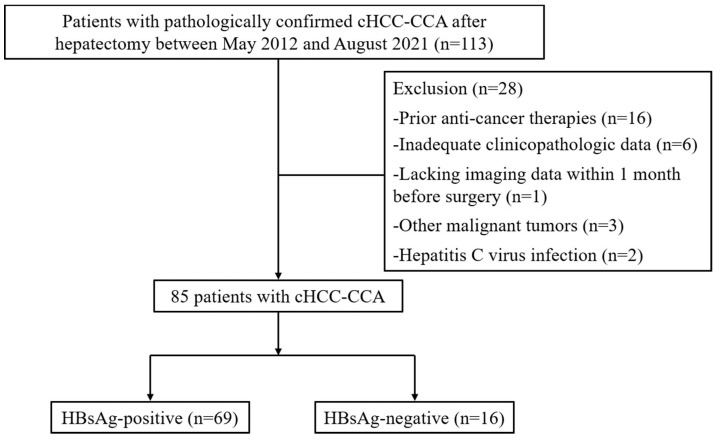
Flowchart of the patient selection process. cHCC-CCA, combined hepatocellular carcinoma and cholangiocarcinoma; MVI, microvascular invasion; HBsAg, Hepatitis B surface antigen.

**Figure 2 jpm-13-00246-f002:**
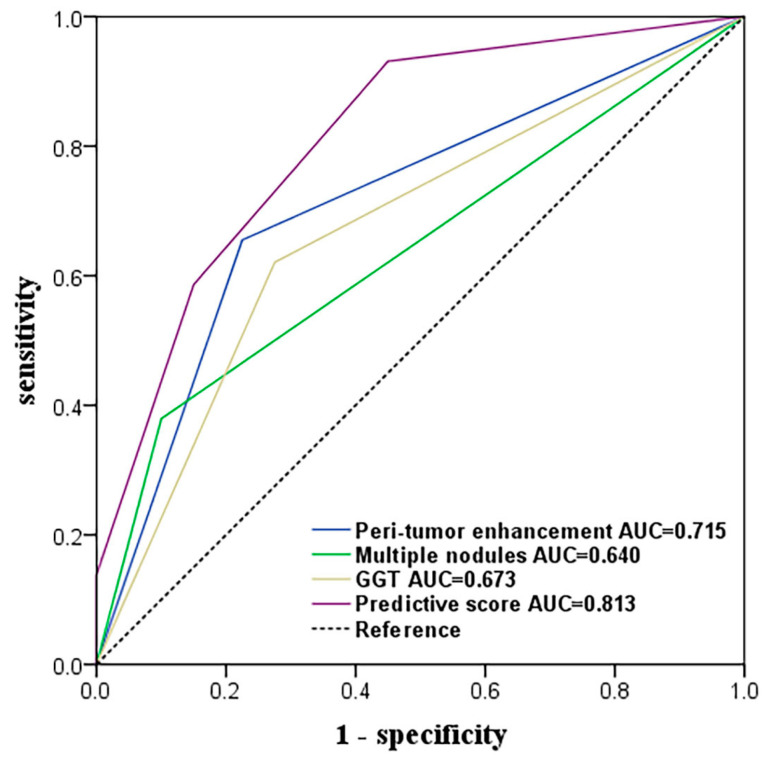
Receiver operating characteristic curves of each independent factor and the prediction score for the prediction of MVI. For the prediction of MVI, the AUC of the prediction score was higher than that of the presence of peri-tumor enhancement (*p* = 0.053), GGT (*p* = 0.004) and multiple nodules (*p* < 0.001).

**Figure 3 jpm-13-00246-f003:**
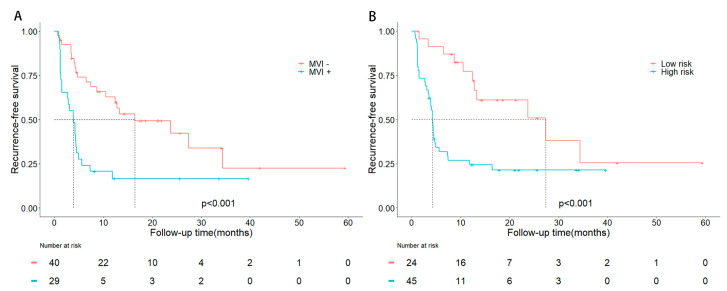
Kaplan-Meier curves of recurrence-free survival stratified by histologic MVI status (**A**) and predicted MVI status (**B**).

**Table 1 jpm-13-00246-t001:** Clinical and imaging characteristics of HBV-related cHCC-CCA patients according to MVI.

Variables	MVI-Negative (N = 40)	MVI-Positive (N = 29)	*p* Value
Age ^a^	51.1 ± 10.2	50.3 ± 8.0	0.702
Gender (male)	33 (82.5)	23 (79.3)	0.982
HBcAb ^b^	38 (95.0)	29 (100)	0.506
HBeAg	7 (17.5)	5 (17.2)	1.000
HBeAb	30 (75.0)	23 (79.3)	0.897
PLT < 100 × 10^9^/L ^b^	2 (5.0)	0 (0.0)	0.506
ALT > 50 (U/L)	4 (10.0)	6 (20.7)	0.302
GGT > 60 (U/L)	11 (27.5)	18 (62.1)	0.009
TBIL > 20 (umol/L) ^b^	3 (7.5)	2 (6.9)	1.000
Alb < 40 (g/L)	5 (12.5)	5 (17.2)	0.837
AFP ≥ 400 (ng/mL)	10 (25.0)	5 (17.2)	0.634
CA 19-9 ≥ 35 (U/mL)	11 (27.5)	14 (48.3)	0.129
CEA ≥ 5 (ng/mL)	8 (20.0)	5 (17.2)	1.000
Tumor size (≥5 mm)	14 (35.0)	18 (62.1)	0.048
Multiple nodules	4 (10.0)	11 (37.9)	0.013
Tumor shape			0.526
Globular	11 (27.5)	5 (17.2)	
Lobulated	19 (47.5)	14 (48.3)	
Irregular	10 (25.0)	10 (34.5)	
Rim enhancement	19 (47.5)	19 (65.5)	0.215
Peri-tumor enhancement	9 (22.5)	19 (65.5)	0.001
Wash out	22 (55.0)	12 (41.4)	0.383
Delayed central enhancement	7 (17.5)	9 (31.0)	0.305
Peritumoral bile duct dilatation ^b^	2 (5.0)	3 (10.3)	0.643
Enhancing capsule ^b^	5 (12.5)	2 (6.9)	0.690
LR-M	23 (57.5)	21 (72.4)	0.308

^a^ Data are expressed as mean ± standard deviation. ^b^ Data were compared using Fisher’s exact test. HBV, hepatitis B virus; HCV, hepatitis C virus; NLR, neutrophil-to-lymphocyte ratio; PLT, platelet; ALT, alanine aminotransferase; GGT, gamma-glutamyl transferase; TBIL, total bilirubin; Alb, albumin; AFP, alpha fetoprotein; CA19-9, carbohydrate antigen 19-9; CEA, carcinoembryonic antigen.

**Table 2 jpm-13-00246-t002:** Univariate and multivariate analyses of risk factors for the MVI of HBV-related cHCC-CCA.

Variables	Univariate Analyses			Multivariate Analyses				Score
	OR	95%CI	*p* Value	β	OR	95%CI	*p* Value	
GGT > 60 (U/L) Tumor size (≥5 mm)	4.31 3.04	1.55–11.99	0.005	1.305	3.69	1.10–12.37	0.034	1
		1.13–8.20	0.028					
Multiple nodules	5.50	1.53–17.71	0.009	1.483	4.41	1.06–18.39	0.042	1
Peri-tumor enhancement	6.54	2.25–19.01	<0.001	1.818	6.16	1.78–21.36	0.004	1

MVI, microvascular invasion; OR, odds ratio; CI, confidence interval; β, partial regression coefficient; GGT, gamma-glutamyl transferase; CA19-9, carbohydrate antigen 19-9.

## Data Availability

The data presented in this study are available on request from the corresponding author.

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
