# Peer review of "Combining Preoperative Clinical and Imaging Characteristics to Predict MVI in Hepatitis B Virus-Related Combined Hepatocellular Carcinoma and Cholangiocarcinoma"

_jpm, 2023, doi:10.3390/jpm13020246_

Round 1

Reviewer 1 Report

Combined hepatocellular carcinoma and cholangiocarcinoma (cHCC-CCA) is a rare form of primary liver malignancy in which microvascular invasion (MVI) is associated with poor postoperative prognosis. This study is a single-center, retrospective study of predictors of MVI in patients with hepatitis B virus (HBV)-associated cHCC-CCA. The authors analyzed 69 patients with HBV-associated cHCC-CCA and found that high serum GGT, peritumoral enhancement, and multinodularity were independent variables associated with MVI. A prognostic model was further constructed and a prediction score of ROC 0.813 (95% CI 0.717-0.908) was achieved. The recurrence-free survival rate was significantly lower in the high-risk group with a score of 1 or higher (p < 0.001). This score may be useful in identifying cHCC-CCA patients who would benefit from surgery.

Major problems:

The analysis is easy to understand and will be useful for treatment strategies for cHCC-CCA patients.

It would be interesting to see if the treatment status and viral load of nucleic acid analogs for hepatitis B affect the analysis.

Miner problem:

Identifying Survival in the Methods section as recurrence-free survival would be easier for the reader to understand.

If possible, an analysis of the validation cohort would be even better.

Reviewer 2 Report

The manuscript entitled “Combining Preoperative Clinical and Imaging Characteristics to Predict MVI in Hepatitis B Virus-related Combined Hepatocellular Carcinoma and Cholangiocarcinoma” by Si-Si Huang , Meng-Xuan Zuo and Chuan-Miao Xie is in principle a well written review that wants to investigate in preoperative predictors of MVI in hepatitis B virus-positive cHCC-CCA patients.

There are minor aspects that could be improved:

1)     Line 134: Please indicate goodness-of-fit (R2-)-values for univariate and multivariate analysis

2)     Line 141: Explain in simple terms what the prediction score represents, how is it calculated? I recommend to discuss the value 0.81 (line 217, limitations), what is “merely” a good value, but not excellent

3)     Line 140/Line 146: term “predictive” score / “prediction” score

4)     Line 198: The authors report that several risk factors including tumor size, tumor number and peritumoral enhancement have already been related with invasiveness of cHCC-CCA. Therefore I recommend to highlight their findings about a GGT-dependency (new aspect) in the discussion section, it should be discussed in more detail, f.i. what are the reported coherences between GGT and aggressiveness (line 198) in literature

5)     Line 220/221: I would suggest “which is not as reliable as…”, further I am not sure what the authors mean by “anti-viral treatment was not included either”. I would recommend to omit this.

Round 2

Reviewer 1 Report

The authors responded appropriately to the reviewers' comments.

As a result, the revised version was deemed Acceptable.